# Evidence for the Transcription of a Satellite DNA Widely Found in Frogs

**DOI:** 10.3390/genes15121572

**Published:** 2024-12-05

**Authors:** Jennifer Nunes Pompeo, Kaleb Pretto Gatto, Diego Baldo, Luciana Bolsoni Lourenço

**Affiliations:** 1Laboratório de Estudos Cromossômicos, Instituto de Biologia, Universidade de Campinas, Campinas 13083-862, SP, Brazil; jennifernunespompeo@gmail.com; 2Laboratório de Citogenética Evolutiva e Conservação Animal, Departamento de Genética, Setor de Ciências Biológicas, Universidade Federal do Paraná, Curitiba 81531-980, PR, Brazil; kaleb.gatto@gmail.com; 3Laboratorio de Genética Evolutiva “Claudio Juan Bidau”, Instituto de Biología Subtropical (CONICET-UNaM), Facultad de Ciencias Exactas Químicas y Naturales, Universidad Nacional de Misiones, Posadas N3300LQF, Misiones, Argentina; diegobaldo@gmail.com

**Keywords:** anura, transcriptome, bioinformatics, genomics

## Abstract

Background: The satellite DNA (satDNA) PcP190 has been identified in multiple frog species from seven phylogenetically distant families within Hyloidea, indicating its broad distribution. This satDNA consists of repeats of approximately 190 bp and exhibits a highly conserved region (CR) of 120 bp, which is similar to the transcribed region of 5S ribosomal DNA (rDNA), and a hypervariable region (HR) that varies in size and nucleotide composition among and within species. Here, to improve our understanding of PcP190 satDNA, we searched for evidence of its transcription in the available transcriptomes of *Rhinella marina* (Bufonidae) and *Engystomops pustulosus* (Leptodactylidae), two phylogenetically distantly related species. Methods: We first characterized the 5S rDNA and PcP190 sequences in these species by searching for them in available genome assemblies. Next, we used the PcP190 (CR and HR) and 5S rDNA sequences of each species as queries to search for these sequences in RNA-seq libraries. Results: We identified two types of 5S rDNA in each analyzed species, with a new type found in *E. pustulosus*. Our results also revealed a novel type of PcP190 sequence in *R. marina* and a new subtype of PcP-1 in *E. pustulosus*. Transcriptome analyses confirmed the expected transcription of the 5S rRNA gene and showed transcription of both the CR and HR of the PcP190 satDNA in both species and in different tissues. Conclusions: As the entire repeat of this satDNA is susceptible to transcription, the high variability observed in the HR cannot be attributed to transcriptional activity confined to the CR.

## 1. Introduction

Satellite DNAs (satDNAs) are characterized by arrays of tandemly repeated units commonly found in heterochromatin domains, such as centromeric and pericentromeric regions [1,2,3]. Many families of satDNA sequences may be categorized as either species-specific or genus-specific [1,4,5], which reflects the rapid evolution of such sequences. The repeat units that compose a family of satDNA may evolve in concert due to diverse mechanisms of nonreciprocal transfer, which favors homogenization throughout the members of a satDNA family and the fixation of a sequence variant within a group of reproductively linked organisms [1,6,7]. This nonindependent evolution of the satDNA sequences can explain the consistent traits that these sequences typically display within species/populations and the higher levels of diversity that exist between species/genera [6,7]. Another important feature of satDNAs is their extraordinary ability to undergo amplification and deletion events, which can lead to variation in the number of repeats of a specific satDNA family among closely related species [1]. This feature is in accordance with the library model [8], which postulates that related species share a collection or library of satDNAs and that species-specific profiles may arise from variations in the copy number of particular satDNA sequences within the shared library.

Although satDNA repeats are noncoding sequences, their transcriptional potential should not be dismissed. Recent studies have examined the transcription of this type of repetitive DNA in various organisms, including insects, nematodes, plants, and vertebrates [9,10,11,12,13]. With advances in sequencing technology and the development of bioinformatic approaches, the study of satDNA became more feasible [5,14]. SatDNA transcripts typically give rise to long noncoding RNAs (lncRNAs) and/or small interfering RNAs (siRNAs), which play roles in processes such as heterochromatin formation and maintenance, centromere identity and formation, kinetochore plate association, and gene expression regulation [5,15].

Given the significance of satDNA among diverse taxa, the study of this type of sequence is very important. For anurans, some satDNAs have been characterized, e.g., [16,17,18,19,20], and one satDNA with a wide taxonomic distribution in this group is PcP190. This satDNA consists of repeat units of approximately 190 bp and has been found in 34 species across seven families of Hyloidea (*sensu* [21]) [22,23,24,25,26,27,28,29,30,31,32,33]. PcP190 satDNA is considered to have originated from 5S ribosomal DNA (rDNA), as a 120 bp segment of its repeat is similar to that of the 5S rRNA gene [31]. This 120 bp segment from the PcP190 satDNA is highly similar among all of the species (~80% similarity [30]), known as the conserved region (CR). In addition to the CR, PcP190 satDNA possesses a hypervariable region (HR) that differs in nucleotide composition and size, both among species and within the same species [26,29,30,32]. Numerous PcP190 sequences have been identified, resulting in the current classification of 14 distinct types of PcP190 satDNA, primarily based on the HR [25,26,29,30].

The PcP190 satDNA was mapped to the karyotypes of diverse species using fluorescent in situ hybridization (FISH), revealing a variable number of clusters in centromeric/pericentromeric heterochromatin. Exceptions were found in *Cycloramphus bolitoglossus*, which showed PcP190 clusters in terminal heterochromatin [22], and *Pseudis tocantins*, with PcP190 clusters in the interstitial heterochromatin of the W chromosome [25]. Notably, PcP190 clusters occurred differentially between the Z and W chromosomes of *Physalaemus ephippifer* [32], *Ps. tocantins*, *Ps. bolbodactyla*, and *Pseudis* sp. [25,26,27]. Given the extensive presence of PcP190 satDNA across Hyloidea, with chromosomal clusters occurring in heterochromatin regions, it was previously hypothesized that this satDNA may play a functional role in the genomes of these anurans [25,30,32]. However, the role of this widespread satDNA remains to be established.

The presence of juxtaposed conserved and variable regions in PcP190 raises the hypothesis that differential selective pressures act on this satDNA. Alternatively, sporadic recombination between PcP190 and 5S rDNA has been evidenced [30] and could account for the highly variable region of PcP190, as different 5S rDNA nontranscribing spacers (NTSs) may have been inserted into this satDNA. Here, we aimed to deepen our understanding of PcP190 satDNA by searching for evidence of its transcription and assessing the hypothesis of differential expression of its CR and HR. We conducted our searches in the available transcriptomes of *Rhinella marina* (Bufonidae) and *Engystomops pustulosus* (Leptodactylidae), which are two phylogenetically distantly related anuran species. Given the similarity between the 5S rRNA gene and CR of the PcP190 satDNA, to enable a thorough analysis, we first characterized the PcP190 sequences and the 5S rDNA of these species.

## 2. Materials and Methods

Since 5S rDNA and PcP190 are similar in nucleotide sequence, and multiple types of both classes of repetitive sequences have been found in several species [30], we first characterized the 5S rDNA and PcP190 sequences in the genomes of *E. pustulosus* and *R. marina*. Next, we used the 5S rDNA and PcP190 sequences as queries to search for evidence of transcription of these repetitive DNAs in RNA-seq libraries from both species. We detail each of these analyses in the following sections.

### 2.1. Characterization of the satDNA PcP190 and 5S rDNA of E. pustulosus and R. marina

We conducted BLAST (Basic Local Alignment Search Tool) v.2.15.0 [34] searches in the genomes of *E. pustulosus* and *R. marina* (available from NCBI: GCA_019512145.1 and GCA_900303285.1, respectively) using the CR of a PcP190 satDNA sequence from *Physalaemus cuvieri* (available from NCBI: JF281109.1) as a query. The PcP190 sequence used as a query was classified as a type I PcP190 sequence, the most common type of this satDNA, which has previously been found even in the *Engystomops* genus [29]. As the CR is highly similar across all species and types of PcP190 (average similarity: 80%), the CR of type I PcP190 has been efficiently used to search for this satDNA in various genomes, having already allowed the identification of PcP190 sequences in the genome of *R. marina* in a previous study [30].

We analyzed the contigs displaying similar sequences to the PcP190 satDNA in the Tandem Repeat Finder (TRF) software v.4.09.1 [35] to generate consensus sequences that represent the PcP190 satDNA of both species. Then, we used these consensus sequences as queries in a new BLAST analysis to refine our searches for PcP190 sequences of each species. These contigs revealed the location of the PcP190 satellite DNA repeats in the genome of the studied species.

For the analysis of the 5S rDNA sequences of these species, we conducted BLAST searches utilizing the transcribing region of the 5S rDNA of *Xenopus borealis* as a query (available on NCBI: V01426). The outcomes of the search, along with genome annotations available on NCBI (GCA_900303285.1 and GCA_019512145.1), enabled us to identify the scaffolds/contigs containing the 5S rDNA and their subsequent extraction from the genome assemblies.

We extracted the sequences of interest from the genome assemblies using the SAMtools Faidx tool v.1.21 [36] and included them in matrices containing all of the sequences previously annotated as PcP190 satDNA and 5S rDNA sequences of anuran species available in public databases (Appendix A). Furthermore, we created a combined matrix with these two repetitive DNAs. We used ClustalW software v.2.1 [37] to align the sequences in each matrix and revised the results manually to exclude partial sequences (i.e., sequences that do not represent entire repeats). Considering that the genome assembling algorithms tend to underestimate the number of identical repetitive sequences [38,39], we performed these analyses to assess the nucleotide sequence rather than the number of the 5S rDNA and PcP190 repeats in the genomes under study.

We categorized the PcP190 satDNA sequences based on the CR and HR size and nucleotide composition, while the 5S rDNA sequences were categorized according to their NTS using the same method. We conducted similarity analyses using MEGA XI v.11.0.13 [40] to calculate p-distances. In this analysis, the alignment gaps were neglected in the pairwise analyses, and N was not treated as a distinct nucleotide. We compared the entire monomer of the PcP190 satDNA, the PcP190 CR, the PcP190 HR, the transcribed region of the 5S rDNA, and the 5S rDNA NTS. We used DnaSP v. 6.12.03 [41] to identify distinct haplotypes and MEGA XI [40] to conduct maximum likelihood analyses using the evolutionary models inferred by the same program as the most suitable for the datasets under analysis (i.e., Kimura-2-parameter with gamma distribution for the analysis of the dataset composed of the PcP190 CR and the transcribed region of the 5S rDNA and Tamura 92 for the analysis of the type I PcP190 HR).

#### Chromosome Mapping of PcP190 Sequences in *R. marina*

Since our searches for PcP190 sequences in the genome assembly of *R. marina* revealed the presence of this repetitive DNA in contigs that were not assigned to any specific chromosome (see Section 3.3), we performed FISH using PcP190 sequences as probes. For comparative purposes, we also mapped the 5S rDNA sequences, which were assigned to chromosome 5 in the genome assembly analysis (see Section 3.2).

We used chromosome preparations and DNA samples of a male specimen of *R. marina* from Nangaritza, Zamora Chinchipe province, Ecuador, housed in Museo de Zoología, Universidad Técnica Particular de Loja, Loja, Ecuador (MUTPL 0256). Chromosome preparations were obtained from intestine, following Schmid et al. [42].

We extracted the genomic DNA from liver fragments according to Medeiros et al. [43]. PcP190 satDNA and 5S rDNA sequences were obtained by PCR from the genomic DNA using the primer pairs P190F (5′-AGACTGGCTGGAATCCCAG-3′)/P190R (5′-AGCTGCTGCGATCTGAC AAGG-3′) [31] (annealing temperature of the primers = 64 °C) and 5S-A (5′-TACGCCCGATCTCGTCCGATC-3′)/5S-B (5′–CAGGCTGGTATGGCCGTAAGC–3′) (annealing temperature of the primers = 63.5 °C) [44], respectively. The resulting fragments were purified via the Wizard SV Gel and PCR Clean-Up System Kit (Promega, Madison, WI, USA) and then subjected to sequencing with the BigDye Terminator Kit (Applied Biosystems, Waltham, MA, USA) following the manufacturer’s instructions. Nucleotide sequences were generated on an automated sequencer using the Human Genome and Stem Cell Research Center/IB-USP sequencing facility. The sequences were edited with the BioEdit Sequence Alignment Editor v.7.2 [45] and aligned with the PcP190 and 5S rDNA sequences obtained from the genome assembly analysis to confirm they corresponded to the sequences of interest.

The PcP190 and 5S rDNA probes were obtained by labeling the amplified fragments with digoxigenin-12-dUTP (Roche) (at a 3:1 ratio of dTTP:labeled dUTP) using a PCR Dig Probe Synthesis Kit (Roche, Basel, Switzerland) and the same primers mentioned above. In each case, the labeled DNA was coprecipitated with sonicated salmon sperm DNA (10 mg/mL) with 3 M sodium acetate (1/10 volume) and ethanol. The pellet was washed with 70% ethanol and then suspended in a hybridization buffer containing 50% formamide, 2× SSC, and 10% dextran sulfate.

We hybridized the probes to chromosome preparations of *R. marina* following the protocol of Viegas-Péquignot [46]. For probe detection, we used an anti-digoxigenin antibody conjugated with rhodamine (Roche) following the manufacturer’s instructions. Chromosomes were stained with DAPI (0.5 μg/mL). Images were captured on an Olympus BX60 fluorescence microscope (Olympus, Tokyo, Japan) and edited only for contrast and brightness using Adobe Photoshop CS4.

### 2.2. Searching for satDNA PcP190 and 5S rDNA Sequences in RNA-seq Libraries of E. pustulosus and R. marina

For this analysis, we mapped reads from the RNA-seq libraries available on NCBI for *E. pustulosus* (PRJNA578590, PRJNA626021) and *R. marina* (PRJNA382870) to the PcP190 satDNA and 5S rDNA consensus sequences of each species.

RNA-seq libraries were obtained from the NCBI-SRA database using the sratoolkit tool v.3.0.1 (https://trace.ncbi.nlm.nih.gov/ accessed on 20 July 2022). We processed the libraries using Trimmomatic software v.0.39 [47] for trimming (following the protocol for paired-end data) and the BWA v.0.7.17 [48] tool for mapping (following the BWA mem algorithm). We analyzed the sorted mapping results using SAMtools to acquire statistical information and map coverage [36]. This information was presented graphically after normalization to log10 using the ggplot2 package from RStudio v.2024.04.0 [49,50,51] and visualized using Tablet software v.1.21.02.18 [52].

To compare the number of transcripts of PcP190 satDNA in the distinct RNA-seq libraries, we utilized the fragments per kilobase of transcript per million mapped reads (FPKM) [53,54] index for normalization. For this analysis, when PE reads were mapped, only one fragment was counted. Bar graphs were created using RStudio software v.2024.04.0 [49,50] and normalized to log2 to illustrate the FPKM values.

## 3. Results

### 3.1. 5S rDNA of E. pustulosus

Four repeat units of the 5S rDNA sequence were identified in the genome assembly of *E. pustulosus*, which were located contiguously in chromosome 5 and corresponded to two types of 5S rDNA. Two of these repeat units had 803 bp/815 bp, of which 120 bp referred to the transcribed region and 683 bp to the NTS (Appendix A). Both sequences were highly similar to each other (94.33%) and showed similarity to the type II 5S rDNA sequences of the Leiuperinae subfamily [30] (Appendix A); therefore, they were classified in this category. The average similarity to the gene region and NTS of all type II 5S rDNA sequences of Leiuperinae (including sequences of *E. pustulosus*) was 93.79% and 74.02%, respectively. The maximum likelihood analysis of the 5S rDNA transcribing regions found in the subfamily Leiuperinae nested these sequences of *E. pustulosus* within the type II 5S rDNA group (Figure 1).

The other two 5S rDNA repeat units found in the *E. pustulosus* genome were 732 bp and 727 bp, with NTSs of 608 bp and 607 bp, respectively (Appendix A). Both repeat units were very similar to each other (99.31%). The gene region of these repeat units was very similar to the type III 5S rDNA of Leiuperinae (Appendix A), which was previously found in *Pleurodema diplolister* [30]. Accordingly, these repeats from *E. pustulosus* and *Pl. diplolister* clustered together in the maximum likelihood analysis of the 5S rDNA transcribing regions from the subfamily Leiuperinae (Figure 1). However, the NTSs of these sequences were only partially similar, since they shared only the first 261 bp (Appendix A). Based on these results, we classified this second type of 5S rDNA found in *E. pustulosus* as a subtype of type III 5S rDNA of Leiuperinae and designated it as type III-b.

### 3.2. 5S rDNA of R. marina

Seven 5S rDNA repeat units were isolated from the *R. marina* genome assembly; these repeat units were found in only one contig of the assembly and were classified into two types. All of these sequences were also found and included in the analysis conducted by Targueta et al. [30].

Three repeat units, classified as type I 5S rDNA of *R. marina*, were 745–752 bp in length and were highly similar to each other (99.3% similarity) (Appendix A). Four additional 5S rDNA repeat units were identified in the genome of this species, all measuring 1086 bp and designated as type II 5S rDNA of *R. marina*. A single sequence differed from the others by only one site, resulting in an average similarity of 99.95% for the four sequences analyzed (Appendix A).

A comparison of the two types of 5S rDNA revealed that their transcribed regions had a similarity of 88.74%, while their NTSs exhibited a similarity of 38.74%.

Fluorescent in situ hybridization with a probe of the 5S rDNA sequence of *R. marina* revealed the presence of these sequences in the terminal region of chromosome pair 5 (Appendix A).

### 3.3. The PcP190 satDNA of E. pustulosus

We found a total of 494 repeat units of PcP190 satDNA in the genome assembly of *E. pustulosus*, which ranged between 171 and 194 bp. These repeat units were present in the contigs assigned to chromosomes 1 to 3 and chromosome 6 and in 10 contigs that were not identified at the chromosomal level.

Among the 494 analyzed sequences, 12 unique haplotypes were recognized (Appendix A). Regarding CR, the average similarity was 86.63%, and the average nucleotide diversity was 0.08228 (SD ± 0.01328). Regarding HR, two distinct sequences, both classified as type I PcP190, were identified. One of these HRs corresponded to the PcP-1a subtype, which is found in various species of Hyloidea [26,29,32], while the other was a new subtype of HR, named PcP-1c. Only one sequence out of the 494 analyzed refers to the PcP-1a subtype, suggesting that this subtype is less abundant in the *E. pustulosus* genome than the PcP-1c subtype. The similarities between the HRs of the PcP-1c sequences and those of the PcP-1a and PcP-1b subtypes were 64.47% and 65.27%, respectively. Maximum likelihood analysis of the HR of all type I PcP190 sequences grouped the PcP-1a sequence of *E. pustulosus* with the PcP-1a sequences from different species (Figure 2).

### 3.4. PcP190 satDNA of R. marina

In the genome of *R. marina*, 46 repeats of PcP190 satDNA were identified in four unmapped contigs. The analysis of these sequences revealed repeat units ranging from 178 bp to 185 bp, with HRs ranging from 63 bp to 65 bp (Figure 3). A total of nine unique haplotypes were identified among the 46 sequences, all corresponding to the same type of PcP190.

The mean similarity of the CR and HR sequences obtained from *R. marina* was 96.88% and 97.47%, respectively. The average nucleotide diversity of CR and HR was 0.01090 (SD ± 0.00359) and 0.01032 (SD ± 0.00417), respectively. The HR found in *R. marina* was not similar to any HR reported previously.

To identify chromosomes of *R. marina* bearing clusters of PcP190 satDNA, we hybridized the chromosomes of *R. marina* with a PcP190 probe constructed from a fragment amplified from the genome of this species. Probe signals were detected in the pericentromeric region of the short arm of chromosome 1 (Appendix A).

### 3.5. Comparisons Between the PcP190 Satellite DNA and the 5S rDNA

The repeat units related to PcP190 satDNA and those related to 5S rDNA were detected on separate contigs within the genome assemblies of the two species investigated in this work.

The transcribed region of all anuran 5S rDNA sequences, including those uncovered in this study, displayed an overall similarity of 82.36%. The CR of all of the PcP190 sequences identified to date, including those detailed in this study, exhibited a similarity of 79.25%. A lower similarity of 69.40% was found between the CR of the PcP190 satDNA sequences and the 5S rDNA transcribing region. Consistent with these findings, the PcP190 CR and 5S rDNA transcribed regions are clearly separated into two distinct groups in the maximum likelihood tree (Figure 4).

### 3.6. Evidence of Transcription of 5S rDNA and PcP190 satDNA

The mapping of reads from the RNA-seq libraries to 5S rDNA sequences provided evidence of transcription for all types of 5S rDNA found in *E. pustulosus* and *R. marina*. The type I 5S rDNA sequence of *R. marina* and the type III-b 5S rDNA sequence of *E. pustulosus* presented a greater number of mapped reads than did the type II 5S rDNA of *E. pustulosus* and type II 5S rDNA of *R. marina* (Appendix A). None of the reads mapped to the 5S rDNA sequences were mapped to the PcP190 satDNA of either species analyzed in this study.

The mapping of reads from the RNA-seq libraries from both *E. pustulosus* and *R. marina* to PcP190 sequences provided evidence of the transcription of PcP190 satDNA. Among the 26 RNA-seq libraries from *E. pustulosus*, 22 provided reads that mapped to the PcP-1c sequence of this species (Figure 5; Appendix A). Four of the 22 RNA-seq libraries returned reads mapped only to the CR of this PcP190 sequence (Appendix A). In the remaining 18 cases, the CR had a greater number of mapped reads than did the HR (Appendix A). The FPKM analysis revealed high variation between the distinct libraries, even between those derived from the same type of tissue (Figure 5B; Appendix A). One library for eye tissue exhibited the highest number of mapped reads and FPKM (Figure 5A, Appendix A). At least one library from each type of tissue displayed the existence of PcP-1c transcripts (Appendix A).

With respect to the PcP-1a sequence of *E. pustulosus*, the less abundant type of PcP190 sequence in the genome of this species (see Item 2.3), reads from all of the RNA-seq libraries analyzed mapped to its CR. None of the reads mapped to the HR of this sequence (Appendix A). The number of reads that mapped to the CR of the PcP-1a sequence was the same as the number that mapped to the CR of the PcP-1c sequence, which could be attributed to the high nucleotide similarity of these CRs (90.08%). After removing the reads that mapped to both the PcP-1a and PcP-1c sequences, only 1–9 reads were exclusively aligned with the PcP-1a sequence. Therefore, even with PcP-1a being less abundant, we could find evidence for the transcription of its CR.

With respect to *R. marina*, the analysis of 13 out of 18 RNA-seq libraries showed evidence of transcription of both the CR and HR of PcP190 satDNA (Appendix A). The CR had a greater number of mapped reads than did the HR (Appendix A), as also observed in the analysis of the *E. pustulosus* transcriptomes. Four other RNA-seq libraries provided reads that mapped exclusively to the CR (Appendix A). The number of mapped reads from the testis RNA-seq library was higher than that from other tissues/organs and tadpole samples (Figure 6A, Appendix A). The FPKM analysis also revealed a much greater value for the testis library [41,49] than for the other libraries of *R. marina*, including the library from the ovary (which had an FPKM of 2.29) (Figure 6B, Appendix A). Unfortunately, the testis is the only organ from male *R. marina* for which transcriptome data are available.

## 4. Discussion

In this study, we identified and characterized the 5S rDNA and PcP190 satDNA repeats of two anuran species that are distantly related phylogenetically and found evidence for the transcription of this satDNA.

To date, 33 species of anurans have had their 5S rDNA characterized. The subfamily Leiuperinae of Leptodactylidae, the most widely studied group, exhibits four distinct types of sequences, with multiple types of 5S rDNA being present in the same species [30]. The variety of 5S rDNA sequences found within the same genome, along with the sharing of some sequence types among different species, implies that the evolution of 5S rDNA in anurans is influenced by both birth-and-death and concerted evolutionary processes [30]. In our study, we also detected two different types of 5S rDNA in both analyzed species, the bufonid *R. marina* and the leptodactylid *E. pustulosus*, which is consistent with previously reported findings in anurans [30].

The two 5S rDNA types found in *R. marina* were the same as those previously found in the analyses conducted by Targueta et al. [30]. For *E. pustulosus*, one of the types of 5S rDNA found was highly similar to the type II sequences of the Leiuperinae subfamily, leading to its classification within this sequence group. The other 5S rDNA sequence, referred to as 5S rDNA III-b in this paper, exhibited similarity with a portion (gene region and first 261 bp of the NTS) of the type III 5S rDNA identified in *Pl. diplolister* by Targueta et al. [30]. However, the remaining segment of the NTS of this *E. pustulosus* sequence exhibited no similarity to any other anuran 5S rDNA type. This suggests that the *Pl. diplolister* and *E. pustulosus* sequences may have shared a common origin but significantly diverged in their NTS, potentially due to a recombination event. Notably, type I 5S rDNA, present in *E. freibergi*, *E. petersi*, and *E.* “*magnus*” [29,55], was not found in our analysis of the *E. pustulosus* genome. Because only four 5S rDNA repeats were recovered from the genome assembly available for *E. pustulosus*, we cannot rule out the possibility that type I 5S rDNA is present in this species but was simply not sampled. On the other hand, considering that *E. pustulosus* is the sister group of a clade that includes *E. freibergi*, *E. petersi*, and *E.* “*magnus*” [56,57], it is also possible that type I 5S rDNA originated after the divergence between both lineages.

The read mapping of RNA-seq libraries indicated that the 5S rDNA sequences we found in both analyzed genomes were expressed. Evidence of 5S rDNA transcription had previously been observed only for type I 5S rDNA sequences of Leiuperinae [30]. Here, we expanded our understanding of Leiuperinae 5S rDNA by providing evidence for the transcription of type II and type III-b 5S rDNA sequences.

PcP190 satDNA has a wide distribution throughout Hyloidea (*sensu* [21]) [22,23,24,25,26,27,28,29,30,31,32,33]. The PcP190 sequence of *R. marina*, previously discovered by Targueta et al. [30], was thoroughly characterized in this study. The sequences of this anuran species revealed a specific HR, which has not been observed in any other species to date, expanding the number of known PcP190 sequence types. There was significant similarity among the copies in both CR and HR (96.88% and 97.47%, respectively), which suggests a high level of homogenization between the repeat units of this satDNA in *R. marina*.

In contrast to that of *R. marina*, the genome of *E. pustulosus* exhibited two different subtypes of PcP190 satDNA sequences. One is the PcP-1a sequence, which is also present in species from the hylid genus *Pseudis* (*sensu* [58]), the leptodactylid genus *Physalaemus*, and in *E. freibergi* [24,25,29,32,33]. The other PcP190 repeats found in *E. pustulosus* refer to a new subtype, PcP-1c. The PcP-1a subtype was much less abundant than the PcP-1c subtype, as only one PcP-1a repeat was found among the 494 PcP190 repeats recovered in the searches performed on the available genome assembly. Regardless of the lower abundance of PcP-1a in *E. pustulosus*, the widespread presence of this subtype in different species from distantly related families suggests that this subtype was already present in the common ancestor of these anurans. The existence of both PcP-1a and PcP-1c, which differ from each other mainly in a particular segment of the HR, may have resulted from recombination involving different HRs or even NTSs of the 5S rDNA since sporadic recombination between these repetitive DNA families has been previously shown [30].

Given that PcP190 satDNA is likely derived from 5S rDNA [31], the study of both families of repetitive sequences requires thorough comparisons. Vittorazzi et al. [31] suggested that the PcP190 satDNA originated from the 5S rDNA based on the similarity (approximately 70%) they noted between the PcP190 region currently known as the CR and the transcribed region of the 5S rDNA of the species in analysis (*Ph. cuvieri*). Considering all of the sequences available to date, including the data we obtained in this work, the average similarity between these two types of sequences is approximately 70%. In addition, the CR and 5S rRNA gene sequences formed two different groups in the likelihood analysis, and none of the described HRs shared any similarity with the known 5S rDNA NTS, supporting PcP190 satDNA and 5S rDNA as two distinct classes of repetitive sequences.

It is known that satDNAs play important roles in eukaryote genomes, including in the formation and maintenance of heterochromatin, the identity and formation of centromeres, and expression regulation, most of which is mediated by long noncoding RNA (lncRNA) and/or small interfering RNA (siRNA) transcripts [5,15,59,60]. However, studies on the transcription and functions of anuran satDNAs are still very limited. Recently, Guzmán et al. [19] described BamHI-800 satDNA from the genome of the Bufonidae species *Bufo bufo* and found evidence of transcription of this satDNA in numerous species of the Bufonidae family via BLASTn analyses using RNA-seq libraries [19]. Here, our findings suggest that the satDNA PcP190 undergoes transcription in various somatic and germinal tissues of *E. pustulosus* and *R. marina*.

Our analyses revealed that the transcription of PcP190 satDNA is not limited to its CR but also occurs from its HR. Only for the HR of the PcP-1a sequence of *E. pustulosus* did we fail to find evidence of transcription, which may be due to the lower frequency of this sequence in the genome of this species. Therefore, we cannot rule out the possibility of transcription also occurring in the HR of the PcP-1a sequence.

The evidence of transcription for both the CR and HR is a crucial finding in studying the role of PcP190 satDNA, as it allows for comparative analysis of these regions. Since the CR, which is the region that corresponds to the 5S rRNA gene, exhibits a much lower evolutionary rate than the HR, it could be hypothesized that greater selective pressure has acted on the CR due to transcriptional activity restricted to this region. Our findings, however, do not support this hypothesis. As they suggest that the entire repeat of the PcP190 satDNA is susceptible to transcription, the high variability in its HR cannot be attributed to transcriptional activity confined to the CR. In this context, the hypothesis that sporadic recombination between PcP190 and 5S rDNA—supported by previous evidence [30]—inserts different 5S rDNA nontranscribing spacers into this satDNA seems to provide a more plausible explanation for the presence of both an HR and a CR in the PcP190 satDNA. Nevertheless, studies on the biological function of the PcP190 transcripts are still needed and may provide new insights into this issue.

In our analyses of transcriptomes, a great number of reads from the eye library of *E. pustulosus* and the testis library of *R. marina* mapped to PcP190 sequences, suggesting the importance of this satDNA in both somatic and germinal tissues. Another interesting finding is that the number of reads from the testis library of *R. marina* mapped to the PcP190 sequence was markedly greater than those regarding all other analyzed libraries, including the ovary library, which had a similar size to the testis library (approximately 210 million reads). Since no other organs apart from the testis were sampled from male *R. marina* for transcriptome analyses, additional comparisons between males and females are currently not possible.

The intricate interplay between satDNA transcripts and heterochromatin formation, establishment, and maintenance has been demonstrated in diverse organisms [3,58,61]. In *Drosophila*, transcripts from satDNA 1.688 have been shown to play a role in the formation of siRNAs, which aid in the formation of heterochromatin in autosomes [62]. Similarly, siRNAs transcribed from the PRAT and PSUB satDNA in Coleoptera are believed to guide the formation of heterochromatin [15]. Chromosomal mapping of PcP190 sequences onto the karyotypes of 32 species from three anuran families [23,24,26,27,28,29] indicated that this satDNA colocalizes with regions of constitutive heterochromatin identified by C-banding. Therefore, it is possible that PcP190 transcripts are involved in the formation and/or maintenance of heterochromatin in anuran species of Hyloidea. However, how the PcP190 transcripts act and their specific contribution to heterochromatin remain to be further investigated.

## Figures and Tables

**Figure 1 genes-15-01572-f001:**
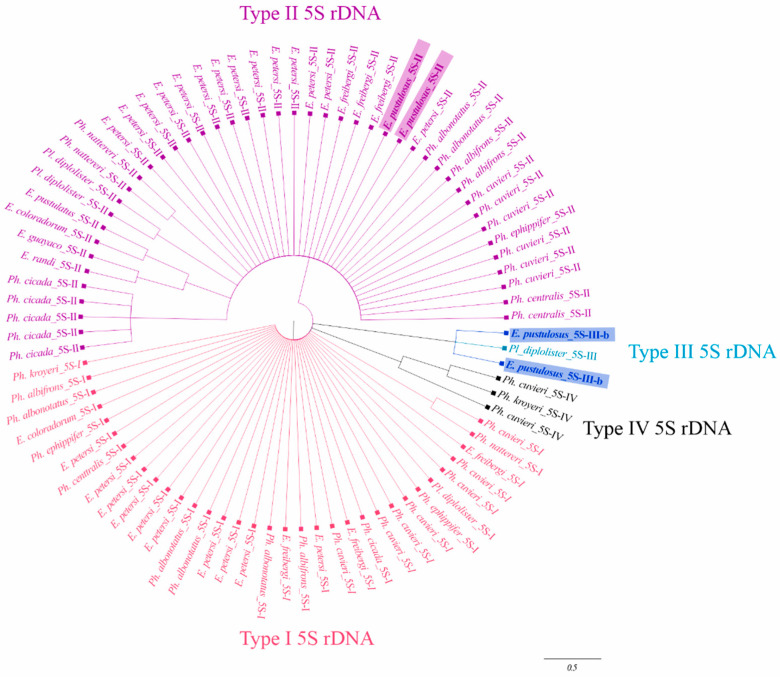
Maximum likelihood analysis of the transcribed region of all types of 5S rDNA found in the subfamily Leiuperinae. Note that the sequences from *E. pustulosus* (highlighted in purple and dark blue) are clustered into two distinct groups, namely type II and type III 5s rDNA.

**Figure 2 genes-15-01572-f002:**
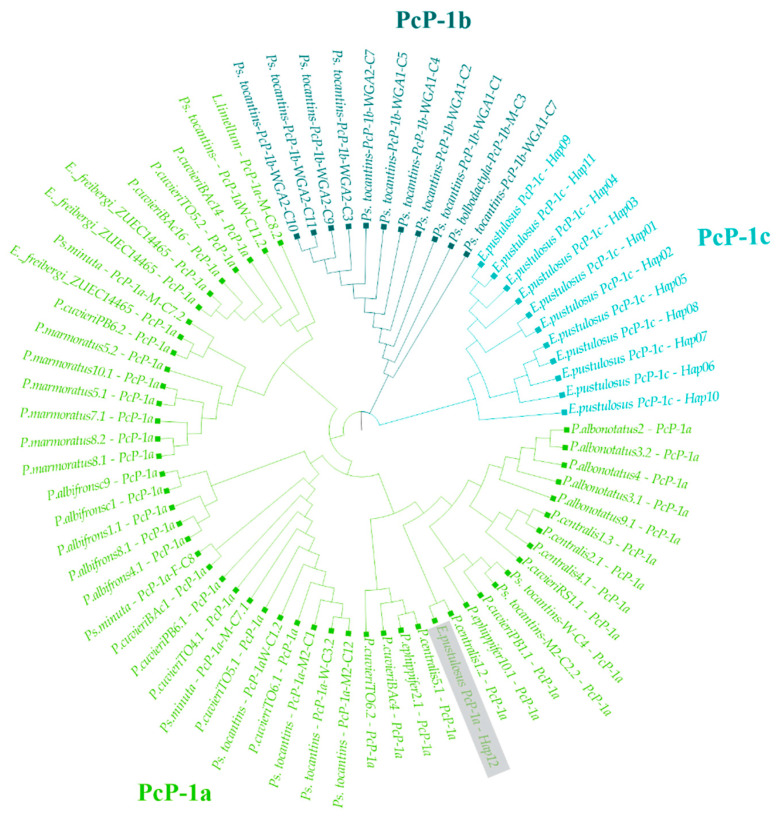
Maximum likelihood analysis of the HR of all type I PcP190 sequences. The branches corresponding to the PcP-1a sequences are shown in light green. The PcP-1a sequence of *E. pustulosus* is highlighted in gray.

**Figure 3 genes-15-01572-f003:**
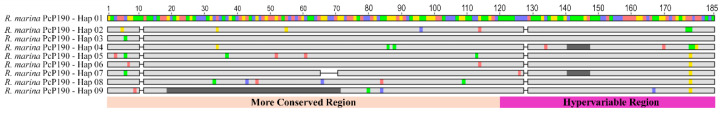
Alignment of the sequences referring to the nine haplotypes representing the PcP190 satDNA found in *R. marina*.

**Figure 4 genes-15-01572-f004:**
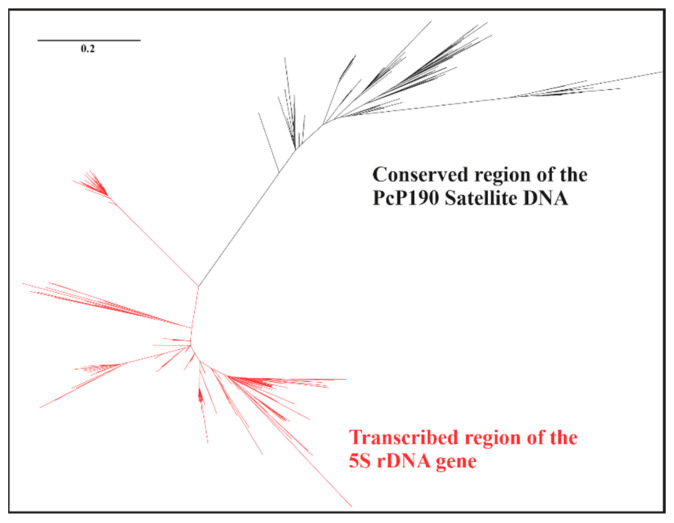
Maximum likelihood analysis of the 5S rDNA transcribing region and CR regions of the PcP190 satDNA of anurans.

**Figure 5 genes-15-01572-f005:**
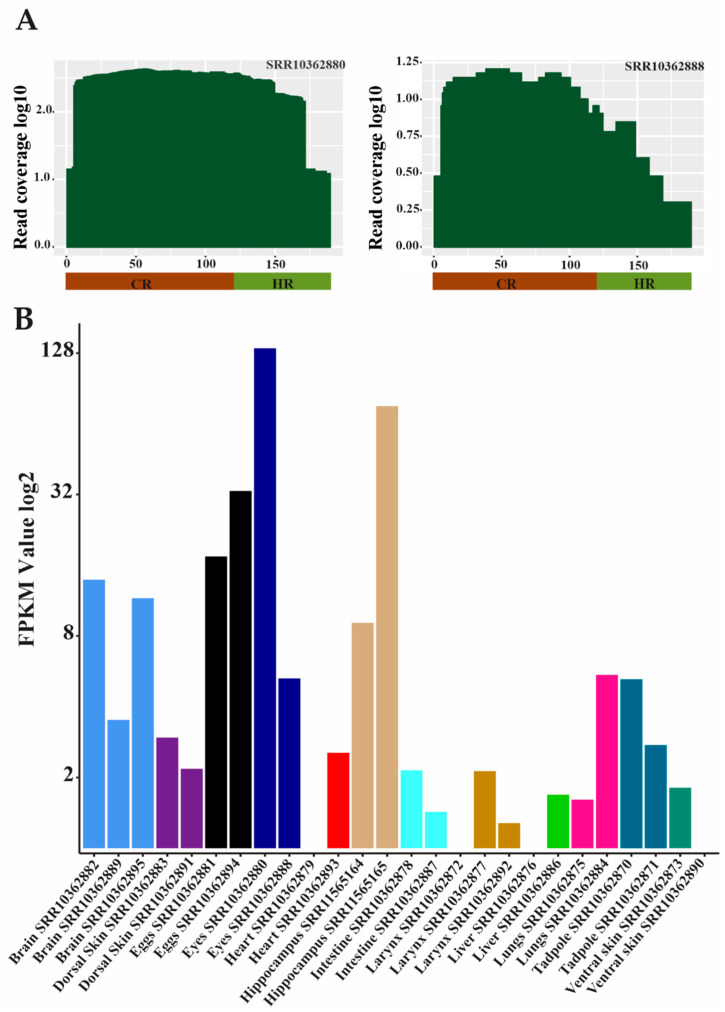
Evidence of the transcription of satDNA PcP-1c in *E. pustulosus*. (**A**) Mapping of reads from two RNA-seq libraries from *E. pustulosus* eyes to the sequence of satDNA PcP-1c from the same species. Note that both the CR (brown bar) and the HR (light green bar) were mapped. (**B**) FPKM values calculated for each library. The accession number of each RNA-seq library is below its respective bar.

**Figure 6 genes-15-01572-f006:**
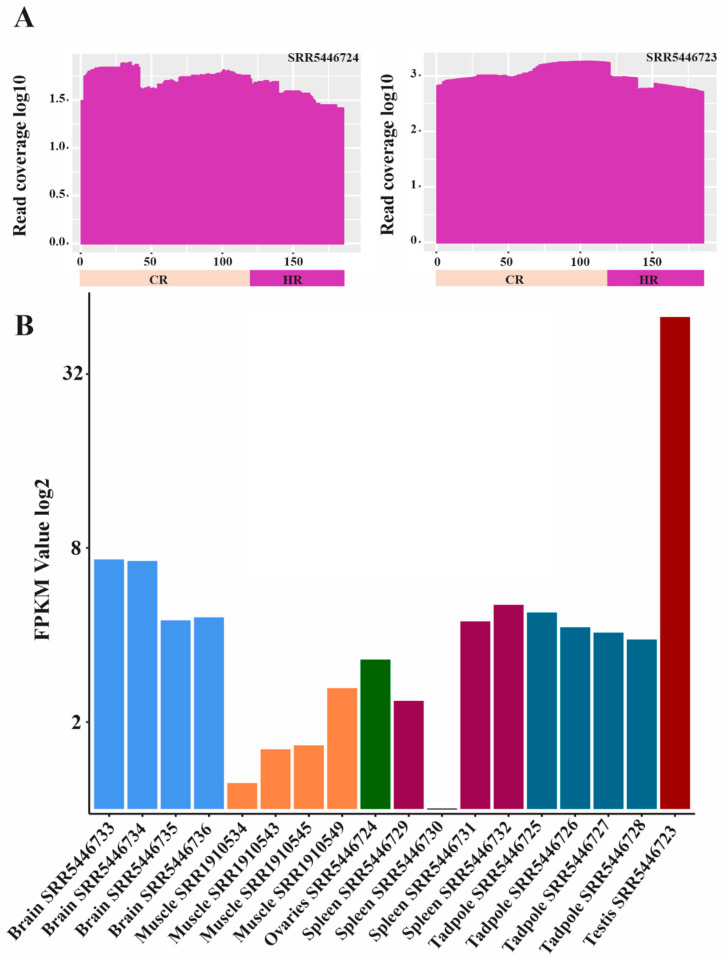
Evidence of the transcription of satDNA PcP190 in *R. marina*. (**A**) Mapping of reads from ovaries (**left**) and testis (**right**) RNA-seq libraries from *R. marina* to a PcP190 sequence from the same species. Note that both the CR (light pink bar) and the HR (dark pink bar) were mapped. (**B**) FPKM values calculated for each library. The accession number of each RNA-seq library is below its respective bar.

## Data Availability

The original contributions presented in this study are included in the article/Appendix A. Further inquiries can be directed to the corresponding author.

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
