# Peer review of "Evidence for the Transcription of a Satellite DNA Widely Found in Frogs"

_genes, 2024, doi:10.3390/genes15121572_

Round 1
Reviewer 1 Report
Comments and Suggestions for Authors
Review 2024- Evidence for the transcription of a satellite DNA widely found in frogs- Pompeo et al
Although previously considered as ‘junk DNA’ due to their non coding nature, satellite DNA are now known to play an integral role in chromosome architecture, including but not limited to centromere, recombination and evolution. Authors Popeo et al studied the distribution of a satDNA called PcP190 in two frog species- Rhinella marina and Engystomops pustulosus. PcP190 has previously been well characterized in thirty four species of Hyloidea. A significant portion of this satDNA (120 bp) is similar to 5S rRNA, suggesting it’s origin. While the manuscript has apt background literature as well as the results based on their analyses, I have the following questions/points to be made-
1. Authors used conserved region (CR) from PcP190 of Physalaemus cuvieri as BLAST query while using the 5S of Xenopus borealis. There is no apparent reasoning as to why these particular genomes were chosen- their phyogeny as related to the subjects/other reasons should be outlined.
2. Lines 118-119- “We conducted similarity analyses using MEGA XI [40] tocalculate p-distances. In this analysis, the alignment gaps were neglected, and N was not treated as a distinct nucleotide.” The reasoning behind ignoring the alignment gaps is not presented, and this may influence monomer size.
3. Supplementary figure S1- Labelling on X axis for species name is blurred and not easy to read in pdf.
4. Similar to FISH mapping for PcP190 and 5S rDNA probes for R.marina, an image should also be provided for E. pustulosus. Additionally, a multicolor FISH with both PcP190 and 5S rDNA probes in different colors (eg Cy3 and Cy5) would enhance the contrasting chromosomal locations in FISH. This would visually enhance/confirm the repeats being present on different contigs as shown by bioinformatic analyses.
5. Line 355- “There was significant similarity among thecopies in both CR and HR (96.88% and 97.47%, respectively), which suggests a high level of homogenization between the repeat units of this satDNA in R. marina.” It would be interesting to assess this for the closest relative of R. marina and ascertain if this is a specific case or similarity between CR and HR region may lead to further recombination between closely related species.
Overall, the manuscript is fairly written with results well explained to the reader.
Author Response
Reviewer #1: Authors used conserved region (CR) from PcP190 of Physalaemus cuvieri as BLAST query while using the 5S of Xenopus borealis. There is no apparent reasoning as to why these particular genomes were chosen- their phyogeny as related to the subjects/other reasons should be outlined.
Authors: The PcP190 sequence used as a query in our study was classified as a type I PcP190 sequence, the most common type of this satDNA, which has previously been found even in the Engystomops genus [Targueta et al., 2018]. As the CR is highly similar across all species and types of PcP190 (average similarity: 80%), the CR of type I PcP190 has been efficiently used to search for this satDNA in various genomes, having already allowed the identification of PcP190 sequences in the genome of R. marina in a previous study [Targueta et al., 2023]. We included this reasoning in Lines 108-117 of the revised manuscript.
The 5S rRNA gene is also a highly conserved sequence, and different sequences would efficiently enable the search of this gene in the genomes of interest. We used a sequence from X. borealis that shares, on average, more than 89% similarity with the known 5S rRNA genes of frogs.
Reviewer #1: Lines 118-119- “We conducted similarity analyses using MEGA XI [40] to calculate p-distances. In this analysis, the alignment gaps were neglected, and N was not treated as a distinct nucleotide.” The reasoning behind ignoring the alignment gaps is not presented, and this may influence monomer size.
Authors: We had only a few alignment gaps and missing data in our dataset, and their impact on the results was not relevant. Nevertheless, in this text, we have explained how MEGA handles gaps and missing data (N) to inform the reader, as other software may use different approaches for treating such cases. By doing so, we aim to ensure the reproducibility of the analyses conducted in our study. Additionally, we emphasize that the sizes of all sequences/monomers identified are reported in the Results section.
Reviewer #1: Supplementary figure S1- Labelling on X axis for species name is blurred and not easy to read in pdf.
Authors: We have replaced this figure in the revised manuscript.
Reviewer #1: Similar to FISH mapping for PcP190 and 5S rDNA probes for R. marina, an image should also be provided for E. pustulosus. Additionally, a multicolor FISH with both PcP190 and 5S rDNA probes in different colors (eg Cy3 and Cy5) would enhance the contrasting chromosomal locations in FISH. This would visually enhance/confirm the repeats being present on different contigs as shown by bioinformatic analyses.
Authors: Although the chromosome mapping of these sequences is certainly interesting, it was not the primary focus of this study. In our analyses of the E. pustulosus genome assembly, some PcP190 sequences we found were assigned to chromosomes 1-3 and 6 (Lines 251-252 in PDF file). In contrast, the PcP190 sequences identified in the R. marina genome assembly were not assigned to any chromosome. Therefore, in this case we performed FISH using PcP-190 probes for this species. For comparative purposes, we also mapped the 5S rDNA probe in chromosomes of this species. We rephrased item 2.1.1 (Lines 153-157 in PDF file) to clarify our rationale.
Due to the limited availability of chromosome preparations, we were unable to perform a double-FISH analysis using both PcP190 and 5S rDNA probes. However, in the revised manuscript, we included two additional metaphase plates hybridized with the 5S rDNA probe. In these plates, chromosome pair 5 (bearing probe signals) can be clearly distinguished from chromosome pair 1 (which lacks probe signals). We believe the updated figure S4 strengthens our findings and helps to dispel any doubts regarding the colocalization of PcP190 and 5S rDNA chromosome clusters.
Reviewer #1: Line 355- “There was significant similarity among the copies in both CR and HR (96.88% and 97.47%, respectively), which suggests a high level of homogenization between the repeat units of this satDNA in R. marina.” It would be interesting to assess this for the closest relative of R. marina and ascertain if this is a specific case or similarity between CR and HR region may lead to further recombination between closely related species.
Authors: This is a very interesting issue. There is no data available for the sister group of R. marina but we could assess this issue in previous studies, based on the PcP190 sequences found in the hylid genera Pseudis and Lysapsus (Gatto et al., 2016, 2018; Targueta et al., 2023). In these genera, several types of PcP190 were found, each one characterized by a typical HR, and we have found evidence of occasional recombination between different types of PcP190 (Gatto et al., 2018) and between PcP190 and 5S rDNA (Targueta et al., 2023).
Cited references
- Gatto, K.P.; Busin, C.S.; Lourenço, L.B. Unraveling the Sex Chromosome Heteromorphism of the Paradoxical Frog Pseudis Tocantins. PLoS ONE 2016, 11, e0156176, doi:10.1371/journal.pone.0156176.
- Gatto, K.P.; Mattos, J.V.; Seger, K.R.; Lourenço, L.B. Sex Chromosome Differentiation in the Frog Genus Pseudis Involves Satellite DNA and Chromosome Rearrangements. Genet. 2018, 9, 301, doi:10.3389/fgene.2018.00301.
- Targueta, C.P.; Gatto, K.P.; Vittorazzi, S.E.; Recco-Pimentel, S.M.; Lourenço, L.B. High Diversity of 5S Ribosomal DNA and Evidence of Recombination with the Satellite DNA PcP190 in Frogs. Gene 2023, 851, 147015, doi:10.1016/j.gene.2022.147015.
Reviewer 2 Report
Comments and Suggestions for Authors
In "Evidence for the transcription of a satellite DNA widely found in frogs", the authors investigated the selective pressures that might be acting on satellite DNA in anurans. They characterized the PcP satellite DNA and 5S rDNA for two species of anurans from transcriptomes of Rhinella marina (cane toad) and Engystomops pustulosus (Túngara frog), and looked for differential expression in the conserved and hypervariable regions and searched for evidence of transcription. They found evidence for transcription in both regions that varied across organs. This study demonstrates an excellent use of publicly available data and the ways it can be used to further biological understanding. The methods seem sound, and the results are important for furthering the understanding of satellite DNA in anurans.
A few small recommendations:
The numbers in Fig S1 are unreadable and the words were hard to read.
Line 56: Typo: "unitsof"
Line 93: Change the order of the NCBI reference numbers. The first refers to the second listed in its current form.
Line 111: What type of manually editing was performed?
Author Response
Reviewer #2: The numbers in Fig S1 are unreadable and the words were hard to read.
Authors: We have replaced this figure in the revised manuscript.
Reviewer #2: Line 56: Typo: "unitsof"
Authors: Corrected.
Reviewer #2: Line 93: Change the order of the NCBI reference numbers. The first refers to the second listed in its current form.
Authors: Done.
Reviewer #2: Line 111: What type of manually editing was performed?
Authors: We used this approach to identify and exclude partial sequences (i.e., sequences that do not represent entire repeats). We included this information in the revised manuscript.
Reviewer 3 Report
Comments and Suggestions for Authors
The manuscript by Pompeo et al. presents an intriguing analysis of the PcP190 satellite DNA (satDNA) in Engystomops pustulosus and Rhinella marina, highlighting the characterization of these genomic elements and the detection of their transcripts. While the study contributes valuable insights into the function of satDNA, the extensive discussion on 5S rDNA within the manuscript raises questions about its relevance to the core findings related to PcP190 satDNA.
1. Relevance of 5S rDNA Discussion: The authors should clarify the purpose and relevance of the extensive discussion on 5S rDNA, especially since they conclude that the transcription of PcP190 satDNA is independent of the 5S rDNA transcription machinery. If the discussion on 5S rDNA is intended to set a comparative context or to rule out potential transcriptional interference or similarities, this rationale should be explicitly stated to help readers understand its inclusion and significance in relation to the main focus of the manuscript.
2. Clarifying the Abstract: The abstract should accurately reflect the content of the manuscript. If 5S rDNA is a major topic of discussion within the paper, it should be appropriately mentioned in the abstract. Conversely, if it is not central to the manuscript’s main conclusions, its presence in the discussion should be scaled back or more tightly integrated with the manuscript’s primary objectives.
Author Response
Reviewer #3: Relevance of 5S rDNA Discussion: The authors should clarify the purpose and relevance of the extensive discussion on 5S rDNA, especially since they conclude that the transcription of PcP190 satDNA is independent of the 5S rDNA transcription machinery. If the discussion on 5S rDNA is intended to set a comparative context or to rule out potential transcriptional interference or similarities, this rationale should be explicitly stated to help readers understand its inclusion and significance in relation to the main focus of the manuscript.
Authors: We provided a detailed comparison between 5S rDNA and PcP190 to support our analysis of transcriptomes and the hypothesis of PcP190 satDNA transcription. Since the 5S rRNA gene and the CR of PcP190 share similarities in their nucleotide sequences, and PcP190 has been proposed to have originated from 5S rDNA, we consider this comparison a crucial component of our study. Given that the nucleotide sequences of 5S rRNA genes are known for only a limited number of anuran species and considering the intriguing evolutionary processes associated with 5S rDNA, we dedicated two (of twelve) paragraphs in the Discussion section to exploring these results. Similarly, we provided an in-depth discussion of PcP190. To highlight the relevance of presenting these data and comparisons, we rephrased the last paragraph of Introduction and included an opening paragraph in the Materials and Methods section.
Reviewer #3: Clarifying the Abstract: The abstract should accurately reflect the content of the manuscript. If 5S rDNA is a major topic of discussion within the paper, it should be appropriately mentioned in the abstract. Conversely, if it is not central to the manuscript’s main conclusions, its presence in the discussion should be scaled back or more tightly integrated with the manuscript’s primary objectives.
Authors: The abstract was rephrased according to these comments.